# Analysis of Bone Histomorphometry in Rat and Guinea Pig Animal Models Subject to Hypoxia

**DOI:** 10.3390/ijms232112742

**Published:** 2022-10-22

**Authors:** Ricardo Usategui-Martín, Álvaro Del Real, José A. Sainz-Aja, Jesús Prieto-Lloret, Elena Olea, Asunción Rocher, Ricardo J. Rigual, José A. Riancho, José Luis Pérez-Castrillón

**Affiliations:** 1Departament of Cell Biology, Genetics, Histology and Pharmacology, Faculty of Medicine, University of Valladolid, 47003 Valladolid, Spain; 2IOBA, University of Valladolid, 47011 Valladolid, Spain; 3Department of Medicine and Psychiatry, Faculty of Medicine, University of Cantabria, IDIVAL, 39011 Santander, Spain; 4Laboratory of Science and Engineering of Materials Division (LADICIM), University of Cantabria, 39011 Santander, Spain; 5Department of Biochemistry and Molecular Biology and Physiology, Faculty of Medicine, University of Valladolid, 47003 Valladolid, Spain; 6Instituto de Biologia y Genetica Molecular (IBGM), University of Valladolid-CSIC, 47003 Valladolid, Spain; 7Department of Nursing, Faculty of Nursing, University of Valladolid, 47003 Valladolid, Spain; 8Internal Medicine Department, Marqués de Valdecilla University Hospital, 39008 Santander, Spain; 9Department of Medicine, Dermatology and Toxicology, Faculty of Medicine, University of Valladolid, 47003 Valladolid, Spain; 10Internal Medicine Department, University Hospital Rio Hortega of Valladolid, 47012 Valladolid, Spain

**Keywords:** hypoxia, bone morphometry, bone remodeling, obesity, micro-computed tomography, animal model

## Abstract

Hypoxia may be associated with alterations in bone remodeling, but the published results are contradictory. The aim of this study was to characterize the bone morphometry changes subject to hypoxia for a better understanding of the bone response to hypoxia and its possible clinical consequences on the bone metabolism. This study analyzed the bone morphometry parameters by micro-computed tomography (μCT) in rat and guinea pig normobaric hypoxia models. Adult male and female Wistar rats were exposed to chronic hypoxia for 7 and 15 days. Additionally, adult male guinea pigs were exposed to chronic hypoxia for 15 days. The results showed that rats exposed to chronic constant and intermittent hypoxic conditions had a worse trabecular and cortical bone health than control rats (under a normoxic condition). Rats under chronic constant hypoxia were associated with a more deteriorated cortical tibia thickness, trabecular femur and tibia bone volume over the total volume (BV/TV), tibia trabecular number (Tb.N), and trabecular femur and tibia bone mineral density (BMD). In the case of chronic intermittent hypoxia, rats subjected to intermittent hypoxia had a lower cortical femur tissue mineral density (TMD), lower trabecular tibia BV/TV, and lower trabecular thickness (Tb.Th) of the tibia and lower tibia Tb.N. The results also showed that obese rats under a hypoxic condition had worse values for the femur and tibia BV/TV, tibia trabecular separation (Tb.Sp), femur and tibia Tb.N, and BMD for the femur and tibia than normoweight rats under a hypoxic condition. In conclusion, hypoxia and obesity may modify bone remodeling, and thus bone microarchitecture, and they might lead to reductions in the bone strength and therefore increase the risk of fragility fracture.

## 1. Introduction

Bone is a highly specialized and dynamic tissue with several fundamental functions. It is crucial for supporting joints, tendons, ligaments, and muscles, it protects vital organs, and it also has key metabolic functions as it a reservoir of calcium and phosphorus [1]. Bone is a very metabolically active tissue and is continuously renewed. This remodeling allows bone to maintain a biomechanical stability and adapt to the changes necessary for the body’s healthy function [1]. Bone remodeling requires a balance between bone formation and bone resorption involving four types of cells: bone lining cells, osteocytes, osteoblasts, and osteoclasts [2].

Sufficient oxygen levels are required to maintain the optimal functioning of the tissue metabolism. The condition in which the metabolism is limited by the absence of sufficient oxygen is called hypoxia. Under hypoxic conditions, cells must perform molecular and physiological adjustments to prolong their survival [3]. The cellular hypoxic response is mediated by hypoxia-inducible factors (HIF), which can modify angiogenesis, glycolysis, programmed cell death, and pH regulation to enhance the oxygen in the cells [4,5]. Evidence suggests that hypoxia may influence bone health [6,7,8]. Although contradictory results have been reported, hypoxia may be associated with decreased osteoblast differentiation and activity and increased osteoclast maturation and activity. Hypoxia has been associated with the inhibition of osteoblast differentiation and maturation mainly due to the alteration of the Runx2, Sox9, Wnt, and PI3K/Akt signaling pathways. In addition, under hypoxic conditions, osteoclastogenesis and osteoclast activity could be increased, mainly through a NF-kB pathway activation and the activation of pro-resorptive genes expression [9]. Contradictory results have also been reported in clinical studies. An exposure to hypoxia has been associated with increased bone fragility in some studies, but it has also been reported that hypoxia is not associated with bone alterations [10,11,12,13]. In this sense, it is important to note that in humans, there is no good model of the effects of chronic hypoxia on the bone metabolism.

It seems clear that hypoxia modifies bone remodeling, but the published results are not uniform and are sometimes contradictory [9]. In this scenario, the aim of this study was to characterize the changes in bone mass subject to various hypoxic conditions for a better understanding of the bone’s response to hypoxia and its possible clinical consequences on the bone metabolism.

## 2. Results

The bone morphometry parameters were evaluated under chronic constant hypoxia and chronic intermittent hypoxia conditions versus normoxia (Figure 1 and Figure 2). In addition, the bone morphometry was analyzed under the conditions of constant and intermittent hypoxia (Table 1) and was compared in hypoxic and normoxic conditions (Appendix A). The influence of obesity was also analyzed (Figure 3). The bone morphometry in hypoxic conditions was evaluated in a guinea pig model (Table 2).

The results showed that chronic constant hypoxia was associated with a more deteriorated cortical tibia thickness, trabecular femur and tibia BV/TV, tibia Tb.N, and trabecular femur and tibia BMD (Figure 1). There were no differences between the rats subjected to constant hypoxia for 7 days or 15 days. In addition, there were no differences in the bone morphometry parameters according to the rat’s sex and age and being subjected to chronic hypoxia. In the case of chronic intermittent hypoxia, the results showed that the rats subjected to intermittent hypoxia had a lower cortical femur TMD, lower trabecular tibia BV/TV, lower Tb.Th of the tibia, and lower tibia Tb.N (Figure 2). There were no significant differences in the analysis of the bone parameters according to age, sex, and time of hypoxia in the rats under chronic intermittent hypoxia. The comparison of the bone morphometry parameters in the states of hypoxia showed no significant differences between the rats subjected to chronic constant hypoxia and chronic intermittent hypoxia (Table 1). Therefore, both conditions were also analyzed globally. The results showed statistically significant differences between the rats subjected to hypoxia and normoxia. The rats who were subjected to hypoxia showed lower values for femur cortical TMD, femur and tibia cortical thickness, femur and tibia trabecular BV/TV, tibia Tb.Th, femur and tibia Tb.N, and trabecular BMD of the femur and tibia (Appendix A). There were no significant differences in the bone morphometry parameters with respect to age, sex, and the time under hypoxia.

The bone morphometry in obese rats who were subjected to hypoxia was also analyzed. The results showed that the obese rats who were subjected to chronic hypoxia had a more deteriorated femur and tibia BV/TV, tibia Tb.Sp, femur and tibia Tb.N, and BMD for the femur and tibia than normoweight rats subjected to hypoxia (Figure 3). There were no significant differences according to age or sex.

Bone morphometry was also analyzed in a guinea pig model who were subjected to hypoxia. The results showed no significant differences between the guinea pigs subjected to hypoxia and those that were not (Table 2).

## 3. Discussion

Bone is a highly dynamic tissue that is continuously renewed through bone remodeling. The purpose of bone remodeling is to maintain the bone biomechanical stability and bone functions [1] and evidence suggests that hypoxia is associated with alterations in bone remodeling, but the published results are contradictory [9]. Therefore, this study analyzed the changes in the bones of rat and guinea pig models who were subjected to hypoxia.

The results showed that the rats exposed to chronic constant hypoxia and chronic intermittent hypoxia had a more deteriorated bone health (in trabecular and cortical bone) than rats who were not. Although the molecular mechanisms associated with chronic constant hypoxia and chronic intermittent hypoxia could be different [14], we also analyzed both conditions globally (chronic constant hypoxia + chronic intermittent hypoxia versus non-hypoxia) and similar results were obtained: the hypoxic status was associated with a more deteriorated bone histomorphometry. It has been reported that hypoxia could reduce osteoblast matrix mineralization and bone formation due to the inhibition of the differentiation and activation of osteoblasts, although the results are contradictory [15,16]. In addition, hypoxia has also been associated with increased osteoclastogenesis and the resorptive capacity of osteoclasts [17,18]. The inhibition of osteoblast differentiation under conditions of hypoxia is associated with a reduced expression of Runx2 and Sp7, factors which are involved in osteoblast differentiation from multipotent mesenchymal cells [19,20,21]. Hypoxia has been associated with increased osteoblast apoptosis due to the inhibition of the PI3K/Akt pathway [22], with an increase in osteoclastogenesis due to NF-kB pathway’s activation [23,24], and with the activation of the expression of the genes involved in the osteoclast fusion and osteoclast resorptive capacity [25,26]. It has also been reported that hypoxia modifies bone remodeling through an altered erythropoietin signaling [27,28,29] and vitamin D metabolism [30]. In this scenario, our results are in line with previous reports which suggested that hypoxia may modify bone remodeling and thus the bone microarchitecture, which may lead to decreased bone strength and an increased risk of fragility fracture associated with hypoxia. On the other hand, our results are in line with previous reports in which bone histomorphometry parameters were analyzed under hypoxic conditions in different rat models [12]. In this sense, it is important to note that our study also includes the histomorphometry analysis of the tibia and that the previous studies used animals exposed to high altitude or hypobaric hypoxia using decompression chambers.

Obesity is one of the main causes of morbidity and mortality in the developed world [31], and the relationship between obesity and bone remodeling is unclear. On the one hand, it has been reported that a high weight and body mass index (BMI) are protective factors against the risk of fracture [32,33]. However, clinical and epidemiological studies have shown that obesity may be a risk factor for fragility fractures [34,35]. Our results showed that obese rats exposed to hypoxia had a more deteriorated bone structure than normoweight rats who were subject to hypoxia. The negative relationship between obesity and bone microarchitecture could be related to the fact that obesity is a proinflammatory state and it has been reported that inflammatory cytokines have a negative influence on bone remodeling. It may also be related to the lower levels of vitamin D associated with obesity which, in turn, could modify the bone metabolism [36,37]. In addition, the association between intermittent hypoxia, obesity, and the oxidative-inflammatory status has also been reported [38]. Therefore, and in this scenario, our results confirm the hypothesis that both hypoxia and obesity are risk factors for a bone microarchitecture modification.

The experiments carried out on guinea pigs showed no significant results. Hypoxia in guinea pigs did not modify the bone morphometry measured by μCT. This may be because guinea pigs (originally from the Andes) could have a reduced ventilatory response to hypoxia, implying a reduced carotid body response to hypoxia and therefore a limited response. Guinea pigs are highly adapted to hypoxia [39].

The main limitation of the study is that we could not determine the relationship between hypoxia, the bone morphometry parameters, and the bone turnover markers. The skeletal effects were only assessed using μCT, thus additional studies could be necessary to validate our findings. Likewise, since this was a short-term study, we do not know what the consequences of longer continuous or intermittent exposures to hypoxia are. In humans, there is no good model of the effects of chronic hypoxia on the bone metabolism and, in addition, there are other associated factors. As there is no good model analyzing the effects of chronic hypoxia in clinical studies, one option could be obstructive sleep apnea syndrome, in which nocturnal hypoxia plays a key role. In this sense, our study may serve as a basis for future studies in patients to help clarify the effect of hypoxia on the bone metabolism.

In conclusion, we found that rats subject to constant and intermittent hypoxia showed worse values for the bone morphometry parameters, probably due to the relationship between the deprivation of adequate oxygen levels and alterations in bone remodeling. In addition, obese rats subject to hypoxia had a worse bone health that normoweight rats under hypoxia, showing that obesity could be considered as another risk factor for bone remodeling. These results are in line with the hypothesis that hypoxia and obesity are risk factors for bone remodeling and thus bone microarchitecture and could increase the risk of reductions in bone strength, therefore increasing the risk of fragility fracture.

## 4. Material and Methods

### 4.1. Animals and Experimental Design

Adult male and female Wistar rats were used. The rats were randomly distributed into experimental groups, 1: chronic exposure to hypoxia for 7 and 15 days (*n* = 26), which was distributed in animals exposed to chronic intermittent hypoxia (*n* = 14) or chronic constant hypoxia (*n* = 12); 2: rats were subjected to normoxia (controls) (*n* = 29); and 3: obese rats were exposed to chronic intermittent hypoxia for 15 days (*n* = 7). The weight did not differ between group 1 and 2 (*p*-value = 0.282). Chronic intermittent hypoxia was characterized by 5% O_2_ for 40 s, then 20% O_2_ for 80 s (30 episodes/h), 8 h/day (from 8:00 to 16:00), for 15 days), as previously described in [40,41]. The rats subjected to chronic constant hypoxia were exposed to 12% O_2_. In addition, male guinea pigs (three months old) were used and divided into controls (*n* = 5) and the animals were exposed to chronic intermittent hypoxia (*n* = 6) for 15 days. The number of experimental units were determined according to previous reports from our group [40,41], and with the aim of complying the principles of the three Rs of experimental animal welfare. All animals had free access to standard food and water and were maintained under the controlled conditions of temperature, humidity, and a stationary light–dark cycle. At the end of the experiments, the animals were euthanized by the administration of a lethal dose of sodium pentobarbital. After euthanasia, the tibias and femurs were collected for further analysis. To avoid a potential significant interference, the data analysis was conducted blindly.

### 4.2. Micro-Computed Tomography (μCT)

The femur and tibia specimens were scanned using a high-energy micro-computed tomography system 527 (SkyScan 11732, Bruker Micro-CT, Kontich, Belgium) and Skyscan 1172 µCT data acquisition software. Since the aim was to maximize the resolution of the samples, the pixel size was reduced to the minimum possible, reaching a pixel size of 6.7 µm and voxel size of 300.76 µm3. Scanning was made at 50 kV and an Al 0.5 mm filter was used to reduce the noise during scanning. During the reconstruction, parameters were used to correct possible beam hardening, ring artifact, and misalignment problems. The maximum and minimum values for the attenuation coefficient were established. The minimum value was set at 0. For the maximum value, the critical section of all scans, the one with the maximum attenuation coefficient value, was selected at the operator’s discretion. Once this section was defined, the maximum value within the histogram was determined and a margin of error of 10% was applied. Finally, the cortical and trabecular areas of the tibia and femur were analyzed. The definition of the histomorphometry area analyzed is showed in the Appendix A. The trabecular bone analysis was made in the distal femur and proximal tibia areas. The regions of interest (ROI) included a total of 3 mm, specifically 2 mm below the growth plate. For the analysis of the cortical area, 3 mm of the central regions of the femur and tibia were selected at 15 mm from the growth plate of the tibia and femur. The ROI delineation for each image was fully automated and assessed, as described by Bruker’s instructions [42,43]. Typical examples of the trabecular and cortical regions are shown in the Appendix A. The scan parameters have been included in the Appendix A.

The structural parameters of the cortical bone which were analyzed were the tissue mineral density (TMD) and cortical thickness. In the trabecular bone, the bone volume over total volume (BV/TV), bone mineral density (BMD), trabecular number (Tb.N), trabecular thickness (Tb.Th), and trabecular separation (Tb.Sp) were analyzed. The TMD represents the volumetric density of the cortical bone, BV/TV indicates the ratio of bone tissue within the whole sample, BMD provides the combined volumetric density of the trabecular bone and soft tissue, and Tb.N, Tb.Th, and Tb.Sp determine the trabecular bone quality.

### 4.3. Statistical Analysis

Continuous variables are expressed as means (standard deviation). The Kolmogorov–Smirnov test was used to analyze the distribution of the continuous variables. In the case of normally distributed variables, the analysis of variance *t*-test was applied. In the case of non-normally distributed variables, the groups were compared using the Mann–Whitney U-test (two groups) or the Kruskal–Wallis test (more than two groups). A *p* < 0.05 was considered significant. All analyses were performed using the SPSS version 22.0 statistical package (SPSS, Chicago, IL, USA).

## Figures and Tables

**Figure 1 ijms-23-12742-f001:**
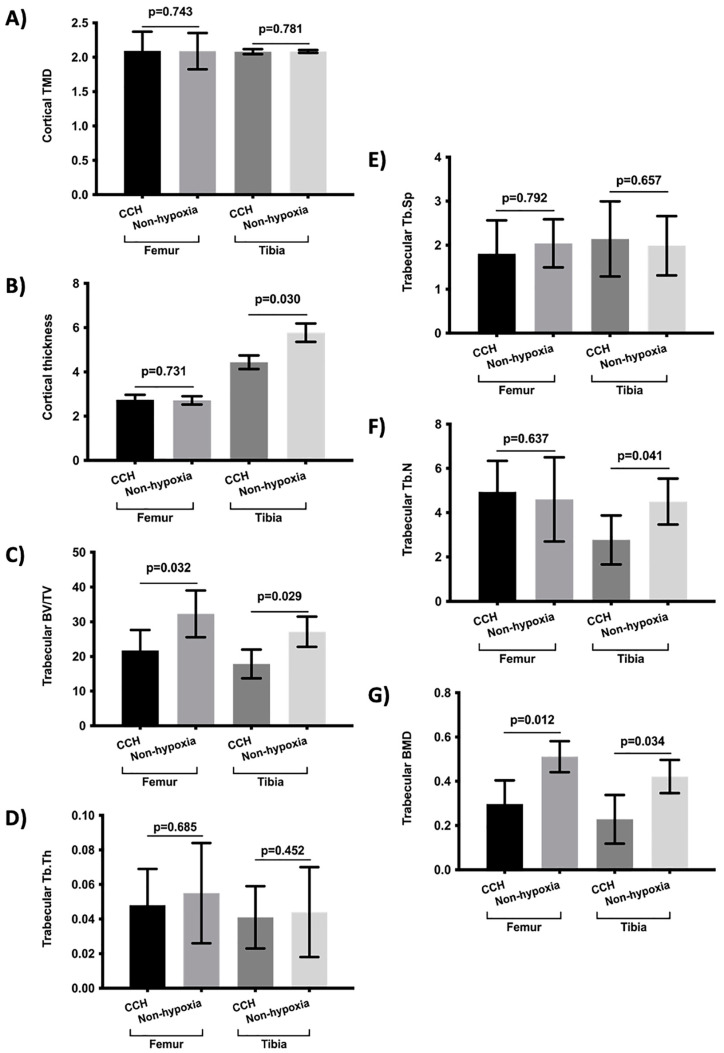
Comparison of bone morphometry parameters between rats subject to chronic constant hypoxic (CCH) condition and controls (non-hypoxia). (**A**) Cortical TMD, (**B**) cortical thickness, (**C**) trabecular BV/TV, (**D**) Tb.Th, (**E**) Tb.Sp, (**F**) Tb.N, and (**G**) trabecular BMD. Variables are presented as mean (standard deviation). TMD: tissue mineral density, BV/TV: percent bone volume; Tb.Th: trabecular thickness; Tb.Sp: trabecular separation; Tb.N: trabecular number; and BMD: bone mineral density.

**Figure 2 ijms-23-12742-f002:**
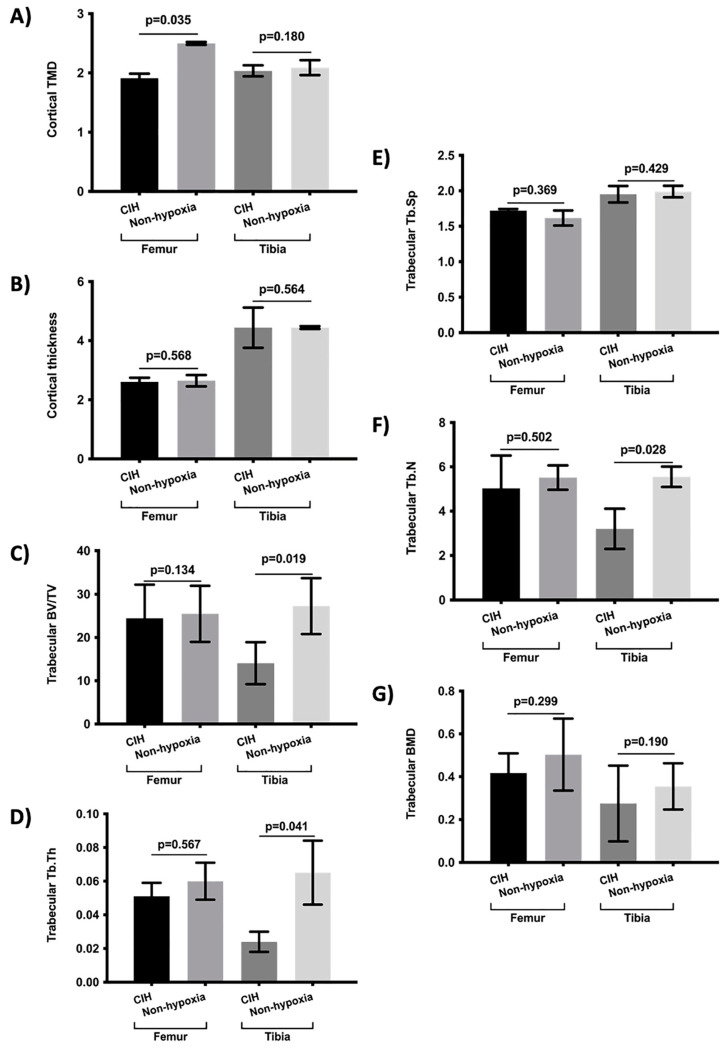
Comparison of bone morphometry parameters between rats subject to chronic intermittent hypoxic (CIH) condition and controls (non-hypoxia). (**A**) Cortical TMD, (**B**) cortical thickness, (**C**) trabecular BV/TV, (**D**) Tb.Th, (**E**) Tb.Sp, (**F**) Tb.N, and (**G**) Trabecular BMD. Variables are presented as mean (standard deviation). TMD: tissue mineral density, BV/TV: percent bone volume; Tb.Th: trabecular thickness; Tb.Sp: trabecular separation; Tb.N: trabecular number; and BMD: bone mineral density.

**Figure 3 ijms-23-12742-f003:**
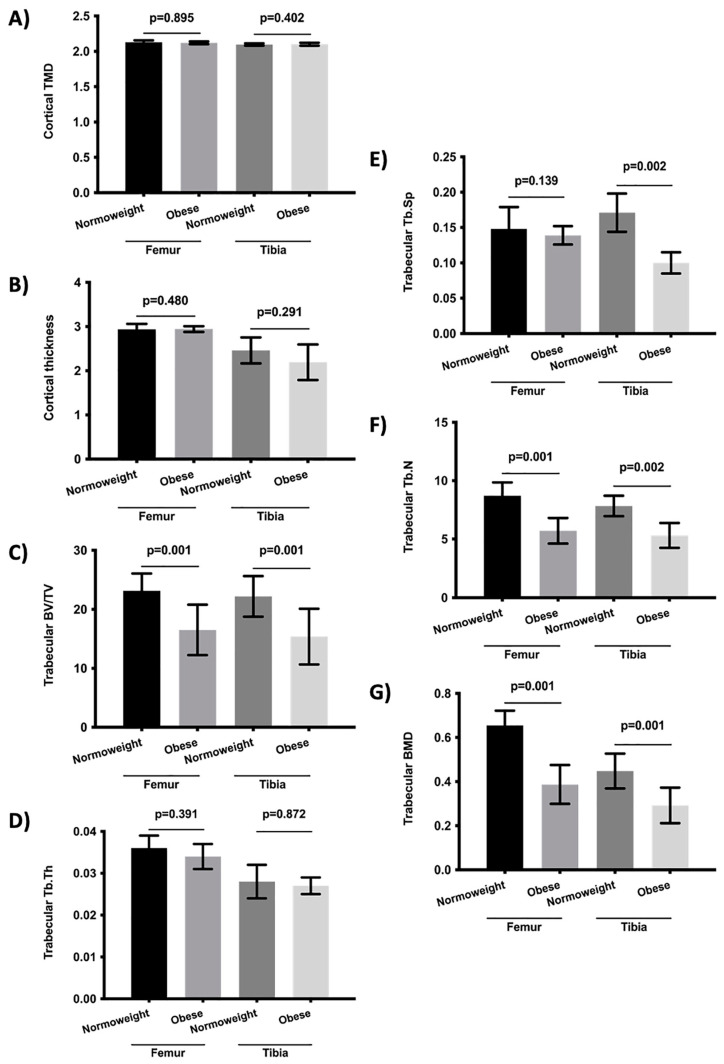
Comparison of bone morphometry parameters between normoweight and obese rats subject to chronic hypoxia. (**A**) Cortical TMD, (**B**) cortical thickness, (**C**) trabecular BV/TV, (**D**) Tb.Th, (**E**) Tb.Sp, (**F**) Tb.N, and (**G**) trabecular BMD. Variables are presented as mean (standard deviation). TMD: tissue mineral density, BV/TV: percent bone volume; Tb.Th: trabecular thickness; Tb.Sp: trabecular separation; Tb.N: trabecular number; and BMD: bone mineral density.

**Table 1 ijms-23-12742-t001:** Comparison of bone morphometry parameters according to different hypoxic conditions.

Bone Parameters			CCH	CIH	Non-Hypoxia	*p*-Value ^1^
Cortical	TMD	Femur	2.092 (0.278)	1.912 (0.075)	2.192 (0.141) *	0.239
		Tibia	2.080 (0.036)	2.035 (0.093)	2.087 (0.074)	0.491
	Thickness	Femur	2.739 (0.225)	2.607 (0.141)	2.681 (0.190)	0.712
		Tibia	4.335 (0.306)	4.440 (0.681)	5.105 (0.231) #	0.842
Trabecular	BV/TV	Femur	21.747 (6.883)	24.429 (7.763)	28.877 (7.112) #	0.089
		Tibia	17.842 (5.160)	18.070 (5.846)	30.194 (5.909) # *	0.101
	Tb.Th	Femur	0.048 (0.021)	0.051 (0.008)	0.050 (0.020)	0.078
		Tibia	0.054 (0.018)	0.041 (0.006)	0.049 (0.023) *	0.198
	Tb.Sp	Femur	1.804 (0.761)	1.720 (0.022)	1.828 (0.327)	0.823
		Tibia	2.142 (0.854)	1.951 (0.716)	1.988 (0.377)	0.389
	Tb.N	Femur	4.935 (1.401)	5.027 (1.485)	5.055 (1.226)	0.506
		Tibia	3.166 (1.896)	3.201 (1.392)	5.025 (1.009) # *	0.319
	BMD	Femur	0.377 (0.127)	0.417 (0.092)	0.437 (0.119) #	0.713
		Tibia	0.228 (0.110)	0.275 (0.177)	0.318 (0.092) #	0.861

Variables are presented as mean (standard deviation). CCH: chronic constant hypoxia, CIH: chronic intermittent hypoxia, TMD: tissue mineral density, BV/TV: percent bone volume; Tb.Th: Trabecular thickness; Tb.Sp: Trabecular separation; Tb.N: Trabecular number; and BMD: bone mineral density. ^1^: *p*-value of the comparison between CCH and CIH, #: *p* < 0.05 between non-hypoxia and CCH, *: *p* < 0.05 between non-hypoxia and CIH.

**Table 2 ijms-23-12742-t002:** Comparison of bone morphometry parameters between guinea pigs subject to hypoxia or not.

Bone Parameters			Hypoxia	Non-Hypoxia	*p*-Value
Cortical	TMD	Femur	2.019 (0.192)	2.145 (0.281)	0.439
		Tibia	1.892 (0.313)	2.023 (0.118)	0.210
	Thickness	Femur	0.612 (0.060)	0.636 (0.129)	0.089
		Tibia	0.560 (0.026)	0.601 (0.293)	0.218
Trabecular	BV/TV	Femur	48.849 (1.292)	49.189 (3.371)	0.591
		Tibia	47.182 (2.032)	48.293 (2.301)	0.189
	Tb.Th	Femur	0.110 (0.012)	0.111 (0.002)	0.742
		Tibia	0.089 (0.006)	0.092 (0.003)	0.087
	Tb.Sp	Femur	0.098 (0.007)	0.121 (0.017)	0.421
		Tibia	0.162 (0.179)	0.129 (0.025)	0.761
	Tb.N	Femur	3.886 (0.419)	4.145 (0.211)	0.139
		Tibia	3.379 (0.388)	3.894 (0.510)	0.201
	BMD	Femur	0.561 (0.132)	0.589 (0.244)	0.459
		Tibia	0.490 (0.198)	0.514 (0.291)	0.278

Variables are presented as mean (standard deviation). TMD: tissue mineral density, BV/TV: percent bone volume; Tb.Th: trabecular thickness; Tb.Sp: trabecular separation; Tb.N: trabecular number; and BMD: bone mineral density.

## Data Availability

Not applicable.

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
