# Peer review of "Analysis of Bone Histomorphometry in Rat and Guinea Pig Animal Models Subject to Hypoxia"

_ijms, 2022, doi:10.3390/ijms232112742_

Round 1

Reviewer 1 Report

The study by Usategio-Martin et al. is interesting and investigates the skeletal effects of chronic and intermittent hypoxia in rats and guinea pigs. The study is interesting and employs μCT to quantify the effects on bone from such exposure. My strongest criticism of the study is only one laboratory methods (μCT) is used. The study would undoubtedly be substantially more solid if more methods were used such as mechanical testing, DXA, bone histomorphometry etc. It is a glaring omission not to include or discuss the findings of a recent review about hypobaric hypoxia (Bone. 2022 Jan;154:116258. doi: 10.1016/j.bone.2021.116258). Additional recent articles about hypobaric hypoxia should also be added.

Below are some specific comments that needs to be addressed:

Abstract:

·      Avoid the term “worse values” to describe the results of the μCT because its meaning is quite opaque. Consider another word choice such as “deteriorated” skeletal integrity.

·      Study duration and sex of animals must be included in the abstract.

·      This sentence should be rephrased for clarity: “Bone morphometry subject to hypoxia and obesity was also analyzed and the results showed that obese rats with hypoxia had worse values for femur and tibia BV/TV, tibia trabecular separation (Tb.Sp), femur and tibia Tb.N and BMD for the femur and tibia than normoweight rats under hypoxic condition.”

·      It must be specified in the abstract that the study used normobaric hypoxia by reducing FiO2.

·      The use of the term “risk factor” is usually reserved to epidemiological studies where it is a determinant associated with an increased risk of a particular disease. It seems a bit out of place to use it in a preclinical study on rodents. For clarity, please rephrase the conclusion.

Introduction:

General comment: The introduction is quite short and superficial and should be expanded with more background information about the preclinical and clinical rationale for conducting the present study.

·      The first paragraph of the introduction is very general and could easily fit into almost all articles about bone. However, is seems a bit unclear and irrelevant to the present study Moreover, why did the authors emphasize the existence of the BMU when it is not mentioned or discussed later in the manuscript? Consider reducing or rephrasing the first paragraph to be more relevant and specific to the main theme about hypoxia.

·      The sentence “In this scenario…” rephrase to clarify what scenario…?

·      The sentence “Although contradictory results have been reported, hypoxia may be associated with decreased osteoblast differentiation and activity and increased osteoclast maturation and activity. Contradictory results have also been reported in clinical studies” needs a proper reference. An obvious choice would be the recent review about preclinical and clinical effects of exposure to hypobaric hypoxia (Bone. 2022 Jan;154:116258. doi: 10.1016/j.bone.2021.116258).

·      Please include a few more and very recent references to add additional support for this sentence “Exposure to hypoxia has been associated with increased bone fragility in some studies, but it has also been reported that hypoxia is not associated with bone alterations [6–9]”. E.g., Front Endocrinol (Lausanne). 2022 Feb 9;13:831369. doi: 10.3389/fendo.2022.831369 and Bone. 2022 Jan;154:116203. doi: 10.1016/j.bone.2021.116203.

·      The paragraph about aim is quite long and would benefit from punctuations to split the sentences. Moreover, how exactly will the present preclinical study in rodents contribute to address the clinical consequences of exposure to hypoxia as stated as one of the aims of the study?

Materials and Methods:

·      Please conform to all necessary reporting from the ARRIVE guideline (see their author checklist: https://arriveguidelines.org/sites/arrive/files/documents/Author%20Checklist%20-%20Full.pdf). Such as item 2b, 5… but please go through the list carefully to ensure everything is reported accordingly.

·      The age of the guinea pigs used should be specified.

·      μCT please specify the (isomtric) voxel size and not only a pixel size.

·      Please clarify what is meant by “…the worst section of all scans was selected at the operator's discretion.”?

·      The delineation of the 3D VOI for analysis must be described in greater details. How was it delineated and what was included? Include the actual length of the VOI for both the trabecular and cortical compartment analyzed. Consider adding a figure to illustrate where on the femur and tibia the analyzes were carried out?

·      Finally, please report all “minimum required” scan parameters as suggested in the guideline for μCT (J Bone Miner Res. 2010 Jul;25(7):1468-86. doi: 10.1002/jbmr.141).

Statistical Analysis:

·      It would also be of interest for the readers to see statistical analyses comparing the effect of CCH and CIH to that of non-hypoxia. This should be done by using a one-way ANOA to distinguish whether the skeletal response is different in rodents subjected to chronic and intermittent hypoxia. Please include such analyses in the manuscript.

Results:

General comment: The graphs are well-organized and easy to comprehend.

Discussion:

·      The first three sentence of the discussion is already (and almost identical) to what is already stated in the introduction – these sentenced could be omitted or should be rephrased.

·      In general, for the manuscript, avoid using the word “worse”. For clarity, use other more precise alternatives.

·      What is meant by globally this statement “we also analysed both conditions globally and similar results were obtained…”?

·      The discussion is lacking an in-depth and detailed discussion about the μCT findings of the present studies and put them in relation to others that have also used μCT (e.g., a list of relevant previous studies utilizing μCT is available in Table 1 in Bone. 2022 Jan;154:116258. doi: 10.1016/j.bone.2021.116258.

·      It should be added to the limitation section that the skeletal effects of hypoxia were only assessed using one modality (μCT). DXA, bone cell analysis, gene expression analysis, bone histomorphometry, or mechanical testing would have strengthened the findings of the study.

Author Response

October, 19th 2022

Dear Sir:

Enclosed please find our revised manuscript by Ricardo Usategui-Martín et al, for consideration for publication in the INTERNATINOAL JOURNAL OF MOLECULAR SCIENCES. The authors greatly appreciate the valuable critiques and comments of the Editor and Reviewers and the opportunity to revise our manuscript. We believe that the revisions based on their recommendations have improved the value and accuracy of our manuscript and we hope that you may consider it suitable for publication in your journal. As requested, we are sending a plain manuscript with accepted changes and a manuscript with tracked changes. The point-by-point answers to editor and reviewers’ comments are provided below.

Yours sincerely,

Prof. José Luis Pérez-Castrillon on behalf of the authors

Faculty of Medicine. University of Valladolid. Valladolid. Spain

Reviewer comments

The study by Usategui-Martin et al. is interesting and investigates the skeletal effects of chronic and intermittent hypoxia in rats and guinea pigs. The study is interesting and employs μCT to quantify the effects on bone from such exposure. My strongest criticism of the study is only one laboratory methods (μCT) is used. The study would undoubtedly be substantially more solid if more methods were used such as mechanical testing, DXA, bone histomorphometry etc. It is a glaring omission not to include or discuss the findings of a recent review about hypobaric hypoxia (Bone. 2022 Jan;154:116258. doi: 10.1016/j.bone.2021.116258). Additional recent articles about hypobaric hypoxia should also be added.

Below are some specific comments that needs to be addressed.

We thank the reviewer for the comments and for the detailed revision of the manuscript. We believe that revisions based on your recommendations have improved the value and accuracy of our manuscript.

Abstract:

1-Avoid the term “worse values” to describe the results of the μCT because its meaning is quite opaque. Consider another word choice such as “deteriorated” skeletal integrity.

We agree with the reviewer, we have avoid the term “worse values”.

2-Study duration and sex of animals must be included in the abstract.

We agree with the reviewer, we have included this information.

3-This sentence should be rephrased for clarity: “Bone morphometry subject to hypoxia and obesity was also analyzed and the results showed that obese rats with hypoxia had worse values for femur and tibia BV/TV, tibia trabecular separation (Tb.Sp), femur and tibia Tb.N and BMD for the femur and tibia than normoweight rats under hypoxic condition.”

We agree with it, we have modified it.

4-It must be specified in the abstract that the study used normobaric hypoxia by reducing FiO2.

We have included that we used normobaric hypoxia models.

5-The use of the term “risk factor” is usually reserved to epidemiological studies where it is a determinant associated with an increased risk of a particular disease. It seems a bit out of place to use it in a preclinical study on rodents. For clarity, please rephrase the conclusion.

We agree with it, we have modified it.

Introduction:

1.General comment: The introduction is quite short and superficial and should be expanded with more background information about the preclinical and clinical rationale for conducting the present study.

We have modified and extended the introduction section.

2.The first paragraph of the introduction is very general and could easily fit into almost all articles about bone. However, is seems a bit unclear and irrelevant to the present study. Moreover, why did the authors emphasize the existence of the BMU when it is not mentioned or discussed later in the manuscript? Consider reducing or rephrasing the first paragraph to be more relevant and specific to the main theme about hypoxia.

We have modified it.

  1. The sentence “In this scenario…” rephrase to clarify what scenario…?

We have modified it.

  1. The sentence “Although contradictory results have been reported, hypoxia may be associated with decreased osteoblast differentiation and activity and increased osteoclast maturation and activity. Contradictory results have also been reported in clinical studies” needs a proper reference. An obvious choice would be the recent review about preclinical and clinical effects of exposure to hypobaric hypoxia (Bone. 2022 Jan;154:116258. doi: 10.1016/j.bone.2021.116258).

We agree with the reviewer, we have included it.

  1. Please include a few more and very recent references to add additional support for this sentence “Exposure to hypoxia has been associated with increased bone fragility in some studies, but it has also been reported that hypoxia is not associated with bone alterations [6–9]”. E.g., Front Endocrinol (Lausanne). 2022 Feb 9;13:831369. doi: 10.3389/fendo.2022.831369 and Bone. 2022 Jan;154:116203. doi: 10.1016/j.bone.2021.116203.

We agree with the reviewer, we have included them.

  1. The paragraph about aim is quite long and would benefit from punctuations to split the sentences. Moreover, how exactly will the present preclinical study in rodents contribute to address the clinical consequences of exposure to hypoxia as stated as one of the aims of the study?

We have shortened the concluding paragraph.

In humans, there is no good model of the effects of chronic hypoxia on bone metabolism and, in addition, there are other associated factors. As there is no good model analyzing the effects of chronic hypoxia in clinical studies, one option could be obstructive sleep apnea syndrome, in which nocturnal hypoxia plays a key role. In this sense, our study may serve as a basis for future studies in patients to help clarify the effect of hypoxia on bone metabolism. We have included it in the discussion section.

Materials and Methods:

1.Please conform to all necessary reporting from the ARRIVE guideline (see their author checklist:https://arriveguidelines.org/sites/arrive/files/documents/Author%20Checklist%20-%20Full.pdf). Such as item 2b, 5… but please go through the list carefully to ensure everything is reported accordingly.

Thank you very much for the comment, we have now included all the missing information.

The experiments were carried out in compliance with applicable international laws and policies (European Union Directive for Protection of Vertebrates Used for Experimental and Other Scientific Ends (2010/63/EU) and were reviewed and approved by the University of Valladolid Institutional Committee for Animal Care and Use.

  1. The age of the guinea pigs used should be specified.

We have included it.

  1. μCT please specify the (isomtric) voxel size and not only a pixel size.

Thank you very much for your comment, the voxel size is 300.76 µm3 and it was added in the text.

  1. Please clarify what is meant by “…the worst section of all scans was selected at the operator's discretion.”?

Indeed, the expression “the worst section” is confusing This sentence was replaced by the following “For the maximum value, the critical section of all scans, the one with the maximum attenuation coefficient value, was selected at the operator's discretion.”

  1. The delineation of the 3D VOI for analysis must be described in greater details. How was it delineated and what was included? Include the actual length of the VOI for both the trabecular and cortical compartment analyzed. Consider adding a figure to illustrate where on the femur and tibia the analyzes were carried out?

We have added new information to explain it more in detail. Also, an illustration has been added as supplemental material to visualize femur and tibia VOI.

  1. Finally, please report all “minimum required” scan parameters as suggested in the guideline for μCT (J Bone Miner Res. 2010 Jul;25(7):1468-86. doi: 10.1002/jbmr.141).

Thank you very much for your comment, it is important in research that the results can be comparable, so it is extremely important to indicate all the scanning parameters. A table with these parameters is attached below and included as supplementary information):

Variable

Standard unit

Value

Voxel size

µm3

300.76

Source Voltage

kV

50

Source Current

µA

100

Exposure time

ms

4920

Frame averaging

N

5

Projections

N

130

  1. It would also be of interest for the readers to see statistical analyses comparing the effect of CCH and CIH to that of non-hypoxia. This should be done by using a one-way ANOVA to distinguish whether the skeletal response is different in rodents subjected to chronic and intermittent hypoxia. Please include such analyses in the manuscript.

Thank you very much for your comment, we have included new column in table 1 with the analysis of the non-hypoxia values and the comparison with CCH and CIH.

Discussion:

1.The first three sentence of the discussion is already (and almost identical) to what is already stated in the introduction – these sentenced could be omitted or should be rephrased.

We agree with the reviewer, we have modified it.

2.In general, for the manuscript, avoid using the word “worse”. For clarity, use other more precise alternatives.

We agree with the reviewer, we have changed it.

3.What is meant by globally this statement “we also analysed both conditions globally and similar results were obtained…”?

We also analysed chronic intermittent hypoxic (CIH)+chronic constant hypoxic (CCH) versus non-hypoxia. The results showed statistically significant differences between rats subject to hypoxia and normoxia. Rats subject to hypoxia showed lower values for femur cortical TMD, femur and tibia cortical thickness, femur and tibia trabecular BV/TV, tibia Tb.Th, femur and tibia Tb.N, and trabecular BMD of the femur and tibia (Supplementary figure 2). We have clarified it.

4.The discussion is lacking an in-depth and detailed discussion about the μCT findings of the present studies and put them in relation to others that have also used μCT (e.g., a list of relevant previous studies utilizing μCT is available in Table 1 in Bone. 2022 Jan;154:116258. doi: 10.1016/j.bone.2021.116258.

Thank you very much for your comment, we agree with the reviewer. We have included a new paragraph in the discussion section.

5.It should be added to the limitation section that the skeletal effects of hypoxia were only assessed using one modality (μCT). DXA, bone cell analysis, gene expression analysis, bone histomorphometry, or mechanical testing would have strengthened the findings of the study

We agree with the reviewer, we have included it in the study limitations section.

Reviewer 2 Report

This article investigated the bone morphometry parameters by micro-computed tomography in rat and guinea pig hypoxia models, and demonstrated hypoxia and obesity may be risk factors for bone remodeling and thus bone microarchitecture.

There were contradictory results in this topic , and this study provided support in bone remodeling by hypoxia.

The conclusions are consistent with the current evidences.

The references are appropriate.

It will be better if the results section can be improved.

In figures, Asterisk could be used to represent the significant difference.

From Figure.1A, there was no significant difference in Cortical TMD. However, in Line 125, “Rats subject to hypoxia showed worse values for femur cortical TMD. At least, the tibia cortical TMD should be also mentioned.

Author Response

October, 19th 2022

Dear Sir:

Enclosed please find our revised manuscript by Ricardo Usategui-Martín et al, for consideration for publication in the INTERNATINOAL JOURNAL OF MOLECULAR SCIENCES. The authors greatly appreciate the valuable critiques and comments of the Editor and Reviewers and the opportunity to revise our manuscript. We believe that the revisions based on their recommendations have improved the value and accuracy of our manuscript and we hope that you may consider it suitable for publication in your journal. As requested, we are sending a plain manuscript with accepted changes and a manuscript with tracked changes. The point-by-point answers to editor and reviewers’ comments are provided below.

Yours sincerely,

Prof. José Luis Pérez-Castrillon on behalf of the authors

Faculty of Medicine. University of Valladolid. Valladolid. Spain

Reviewer comments

This article investigated the bone morphometry parameters by micro-computed tomography in rat and guinea pig hypoxia models, and demonstrated hypoxia and obesity may be risk factors for bone remodeling and thus bone microarchitecture. There were contradictory results in this topic, and this study provided support in bone remodeling by hypoxia. The conclusions are consistent with the current evidences.  The references are appropriate.

We thank the reviewer for the comments and for the detailed revision of the manuscript. We believe that revisions based on your recommendations have improved the value and accuracy of our manuscript.

It will be better if the results section can be improved. In figures, Asterisk could be used to represent the significant difference.

Thank you very much for your comment. we have partially modified the results section. On the other hand and following the recommendations of the other reviewer, we have added a column for non-hypoxia data in table 1. In this table, comparisons that were statistically significant have been marked with asterisk (*or #).

From Figure.1A, there was no significant difference in Cortical TMD. However, in Line 125, “Rats subject to hypoxia showed worse values for femur cortical TMD. At least, the tibia cortical TMD should be also mentioned.

Thank you very much for your comment, we have modified it in the results text.

Round 2

Reviewer 1 Report

My concerns have been adequately addressed.